# Crime Risk and Depression Differentially Relate to Aspects of Sleep in Patients with Major Depression or Social Anxiety

**DOI:** 10.3390/brainsci14010104

**Published:** 2024-01-22

**Authors:** Heide Klumpp, Cope Feurer, Fini Chang, Mary C. Kapella

**Affiliations:** 1Department of Psychiatry, University of Illinois at Chicago, Chicago, IL 60612, USA; feurer@uic.edu (C.F.); finic@uic.edu (F.C.); 2Department of Biobehavioral Nursing Science, University of Illinois at Chicago, Chicago, IL 60612, USA; mkapel1@uic.edu

**Keywords:** sleep, depression, anxiety, geocoding, actigraphy, neighborhood characteristics

## Abstract

Individuals with internalizing conditions such as depression or anxiety are at risk of sleep difficulties. Social–ecological models of sleep health propose factors at the individual (e.g., mental health) and neighborhood (e.g., crime risk) levels that contribute to sleep difficulties. However, these relationships have been under-researched in terms of internalizing conditions. Therefore, the current study comprised participants diagnosed with major depression (*n* = 24) or social anxiety (*n* = 35). Sleep measures included actigraphic variables (i.e., total sleep time, waking after sleep onset, sleep onset latency) and subjective sleep quality. Geocoding was used to assess nationally-normed crime risk exposure at the person level (e.g., murder, assault) and property level (e.g., robbery, burglary). Analyses consisted of independent *t*-tests to evaluate potential differences between diagnostic groups. To examine relationships, multiple regressions were used with internalizing symptoms, crime risk, and age as independent variables and sleep measures as the dependent variable. The *t*-test results revealed that groups differed in symptoms and age but not sleep or neighborhood crime. Regression results revealed crime risk positively corresponded with sleep onset latency but no other sleep measures. Also, only depression positively corresponded with total sleep time. Preliminary findings suggest exposure to crime and depression relate differentially to facets of sleep in individuals with internalizing conditions.

## 1. Introduction

Problematic sleep is widespread among individuals with internalizing conditions such as depressive and anxiety disorders. For example, individuals with major depression frequently experience insomnia (e.g., problems initiating sleep and/or maintaining sleep) [1,2], and longitudinal data indicates there is a causal relationship between insomnia and depression [3]. Moreover, sleep disturbances (e.g., increased sleep onset latency, short sleep duration, impaired sleep maintenance) have been observed to various degrees in different anxiety disorders such as social anxiety disorder, panic disorder, and generalized anxiety disorder [4,5]. In addition to the frequency of co-occurring problematic sleep in internalizing conditions, problematic sleep may contribute to the development and severity of depression and anxiety symptoms [6,7,8]. Altogether, there is evidence that problematic sleep is pervasive and may be a transdiagnostic feature of depressive and anxiety disorders as it cuts across diagnostic categories.

A separate line of research has demonstrated that social determinants are important for sleep health, which has implications for increasing our understanding of sleep in internalizing conditions. For instance, a longitudinal study based on a community sample of low-income neighborhoods showed that exposure to crime rate in the area one resided in was associated with increased risks of insufficient sleep and fragmented sleep (i.e., waking after sleep onset) due, in part, to psychological distress [9]. Findings build on previous research that has shown individuals from disadvantaged neighborhoods, often observed in urban settings, are at an increased risk of insufficient sleep (i.e., less than the recommended seven hours per night) [10,11,12,13]. While the mechanisms remain to be established, they likely involve psychological (e.g., perceived threat) and physical (e.g., environmental noise) factors [11,14,15]. For example, community-level stressors such as neighborhood disorder (i.e., perceived crime, vandalism, noise) [16] may contribute to excessive arousal, which is a key component of insomnia [17]. 

Neighborhood crime may be especially salient among individuals with internalizing conditions as the preferential processing of a threat (e.g., attentional bias to threat-relevant information) is prevalent in anxious, depressed, and vulnerable individuals [18,19]. For example, direct or indirect exposure to crime could affect sleep by contributing to pre-sleep arousal as cognitive and/or somatic hyperarousal negatively impacts sleep [17,20,21,22,23,24]. In a qualitative study regarding sleep in an urban community, a factor that contributed to insufficient sleep was ‘safety insecurity’ such as worrying about getting shot or robbed and concerns when hearing ambulance/police sirens and gunshots [25]. Even though the study did not evaluate pre-sleep arousal, it stands to reason that crime risk may figure into pre-sleep arousal, particularly in light of evidence that crime-related psychological distress negatively affects sleep [9]. While findings point to the role neighborhood stressors may play in problematic sleep, internalizing symptoms are also expected to hinder sleep. For example, symptoms of depression and anxiety include self-critical thoughts, irritability, worry, and somatic symptoms (e.g., heart palpitations, headaches) [26,27], which may impede sleep if they occur around bedtime or promote sleep-interfering behavior (e.g., excessive naps during the day) [28,29]. According to social–ecological models of sleep health, individual-level (e.g., mental health) and neighborhood-level factors (e.g., neighborhood safety) contribute to sleep difficulties [11,30]. However, the extent to which internalizing symptoms (i.e., depression, anxiety) and neighborhood crime relate to sleep uniquely or together remains unclear as this has been under-researched. 

Therefore, the objective of the current study is to examine sleep as it corresponds to internalizing symptoms and crime risk in participants seeking treatment for major depressive disorder or social anxiety disorder, as both are common internalizing conditions. More specifically, the lifetime prevalence rate of major depression in the United States is about 16%, and for social anxiety, it is approximately 12.1% [31]. Common symptoms of major depression include sadness, loss of pleasure, problems with appetite, and a dysregulated activity level (e.g., psychomotor retardation), whereas social anxiety is marked by excessive anxiety and/or avoidance concerning different social and performance situations such as meeting strangers and public speaking [26]. Both disorders are associated with a high burden. For example, a leading cause of disability is major depression [32,33,34], and many individuals with social anxiety experience impairment in developing and maintaining interpersonal relationships and in academic and occupational settings [35,36,37]. Based on previous study findings that indicate that sleep difficulties are transdiagnostic, in that they have been observed in both depressive and anxiety disorders, together with evidence that a greater depression severity and social anxiety severity are associated with more problematic sleep [6,38,39,40], we hypothesized higher levels of depression and higher levels of social anxiety would correspond with worse sleep (e.g., shorter sleep duration, worse sleep quality). We also expected that greater neighborhood crime would correspond with worse sleep. We did not have more specific hypotheses regarding sleep parameters or whether the type of crime would differentially relate to sleep due to the dearth of literature on this topic. 

## 2. Materials and Methods

### 2.1. Participants

The original study used a multimethod approach to determine predictors of psychotherapy outcome and mechanisms of symptom improvement following psychotherapy in unmedicated participants seeking treatment for major depression or social anxiety (ClinicalTrials.gov Identifier: NCT03175068). The current study is based on secondary data analysis as the original study did not have aims that involved sleep or neighborhood characteristics. Consequently, participants were not selected for sleep difficulties or crime risk, and the current study focused on pre-treatment data. Moreover, crime data were only available for participants who provided their addresses and resided in the state of Illinois. Data collection for the current study spanned 4 years (2017 to 2020). The original study was 5 years (2017 to 2021); however, wrist actigraphy data collection was discontinued due to the COVID-19 pandemic. 

All participants were required to be at least 18 years of age and not older than 65 years of age. There were various exclusion criteria, which were based on self-reporting and observation by trained staff. These included, but were not limited to, the following: (1) history of or current psychosis; (2) major medical illness, for example, cancer; (3) plan or intent for suicide or self-harming behavior such as self-cutting or burning; (4) use of psychotropic medication in the last 6 weeks before study entry or during the study; (5) cognitive problems (e.g., traumatic brain injury, dementia); (6) developmental disorders such as autism or learning disability; and (7) substance abuse or dependence in the moderate or severe range 6 months prior to study entry. Comorbid diagnoses were allowed. However, to evaluate the potential unique contribution of major depression and social anxiety to the original study aims, participants with major depression were not permitted to have concurrent social anxiety disorder, and participants with social anxiety disorder were not permitted to have concurrent major depressive disorder. See Table 1 for comorbid diagnoses, which includes insomnia and hypersomnolence. Since the original study did not have sleep aims, a comorbid sleep disorder was neither an inclusion nor exclusion criterion. 

For recruitment, flyers with details of the study were disseminated in the Chicagoland area and social media was also used to advertise the study. Additionally, clinicians who provided treatment for individuals with mood and anxiety disorders in outpatient clinics were informed of the study. Eligibility consisted of a two-step process. First, individuals interested in the study completed a short phone screen. If the phone screen indicated the individual may be eligible for the study, they were invited to complete the consent process and a psychiatric interview that included various clinical measures to determine the individual’s principal diagnosis and comorbid diagnosis. Study eligibility and diagnosis included the input of at least three study staff members, for example, a psychologist, psychiatrist, and social worker (i.e., a best-estimate/consensus panel). The study took place at the University of Illinois in Chicago and the consent form and all other parts of the study received approval from the Institutional Review Board at the University of Illinois in Chicago. Also, all study procedures were compliant with the Declaration of Helsinki. For the participants’ time in the study, participants were compensated. 

### 2.2. Measures

#### 2.2.1. Clinical Measures

After receiving the participant’s consent, participants completed the DSM-5 structured clinical interview [41]. In addition to the structured interview, participants completed the Liebowitz Social Anxiety Scale (LSAS) [42] and Hamilton Depression Rating Scale (HAMD) [27]. These measures were interviewer based and were used to evaluate the severity of social anxiety and depression, respectively. All interviewer-based measures were conducted by trained staff members, who had a minimum of a bachelor’s degree and received individual and group training and supervision by a licensed clinical psychologist with over a decade of experience in assessment. The LSAS consists of 24 items and each item assesses the level of anxiety/fear and level of avoidance regarding various social and performance situations. Internal consistency for the LSAS in the current study was excellent (Cronbach’s alpha = 0.960) [43,44]. The HAMD consists of 17 items that include assessment of sadness, feelings of guilt, psychomotor retardation, loss of pleasure, and somatic symptoms. Internal consistency for the HAMD in the current study was acceptable (Cronbach’s alpha = 0.648) [43,44]. 

When evaluating relationships between depression severity and sleep measures, a modified version of the HAMD was used. Specifically, items 4, 5, and 6 were removed as they assess early, middle, and late insomnia, respectively. This version of the HAMD is hereafter referred to as the ‘modified HAMD’. Internal consistency for the modified HAMD was also acceptable as Cronbach’s alpha was between 0.60 and 0.70 (Cronbach’s alpha = 0.623) [43,44]. 

#### 2.2.2. Sleep Measures

Wrist actigraphy was used to estimate objective sleep. Participants were instructed to wear an actigraph device (30 s epochs; Actiwatch Spectrum, Respironics, Bend, Oregon) on their non-dominant wrist continuously for 7 days and 7 nights. They were also instructed to press the event marker on the device when intending to go to sleep and waking up and to record sleep and wake times in a sleep log, which was used for actigraphy data processing. The sleep log also consisted of binary responses (i.e., yes/no) to three questions that related to sleep problems (e.g., “Did you have difficulty staying asleep?”, “Do you think you have a sleep problem?”). There was also a question regarding removal of the wrist actigraphy device and, if it was removed, the duration it was not worn. Since the primary goal of the sleep log was for processing actigraphy data, responses to the three sleep questions were not entered for analysis in the current study. 

The default settings from the Respironics Actiware 6.0.9 program (10 immobile or mobile minutes for sleep onset or offset and a wake activity count threshold of 40) was used to examine actigraphy data. The default settings were combined with a validated approach to establish the setting of nightly rest intervals, which was informed by event markers, the sleep logs, and light data obtained by the device along with activity levels [45]. This approach was used to examine the following validated sleep variables [45]: total sleep time (TST), which is the number of minutes scored as sleep in each rest interval; waking after sleep onset (WASO), which is the number of minutes of all wake epochs between sleep onset and offset; and sleep onset latency (SOL), which is the number of minutes of all wake epochs from the time in bed to sleep onset. Accordingly, TST reflects sleep duration, WASO represents fragmented sleep, and SOL reflects the time it takes to enter sleep. All variables were scored for each 24 h period and means were computed. 

Subjective sleep was evaluated with the Pittsburgh Sleep Quality Index (PSQI) [46]. The PSQI consists of 19 questions that assess 7 components of sleep (e.g., sleep disturbance, sleep duration, sleep latency) over the period of one month. A global measure of sleep quality is based on the sum of all the components of sleep. The PSQI has been shown to have acceptable psychometric properties [47] and higher PSQI global scores reflect worse sleep quality. A PSQI global score of more than 5 reflects problematic sleep in the clinical range [46].

#### 2.2.3. Neighborhood Crime Exposure

The 2023 CrimeRisk (https://appliedgeographic.com) database [48] was used to obtain indices of neighborhood crime risk at the block group level from participants who resided in Illinois. The database is derived from analysis of past crime reports and provides an assessment of the relative risk of personal (i.e., murder, rape, robbery, assault) and property (i.e., burglary, larceny, motor vehicle theft) crime occurring in one’s neighborhood compared to the national average. Participants’ neighborhoods were defined as the census block group of their current home address at the time of study enrollment. To reduce the number of comparisons, crime risk was based on two measures—personal and property—specifically, the sum of different crimes representing these categories. 

#### 2.2.4. Crime in Illinois

To estimate the rates of crimes more broadly during the data collection period from 2017 to 2020, inclusive, we report the annual index crime reports from the Illinois State Police for these years (Table 2) since the participants in the current study resided in Illinois. The index crime report reflects offenses, arrests, and associated crime rates per 100,000 inhabitants; categories of crime include criminal homicide, rape, robbery, aggravated battery/aggravated assault, burglary, theft, and motor vehicle theft among other crimes (e.g., arson). 

### 2.3. Statistical Analyses

To test for potential differences between diagnostic groups, symptom measures, sleep data, crime risk, and demographic data were submitted to independent *t*-tests and chi-square analyses. 

To evaluate relationships between internalizing symptoms (i.e., depression, social anxiety) and sleep in addition to crime risk and sleep, a series of Pearson correlations were performed. To adjust for multiple comparisons, Bonferroni correction was applied for the two symptom measures (i.e., depression and social anxiety) and the two crime risk measures (personal and property). Specifically, when evaluating depression–sleep relationships and social anxiety–sleep relationships, *p* = 0.025 (0.05/2) denoted significance. The same approach was used when evaluating personal and property crime risk as they pertain to sleep. In light of the novelty of the study, we elected not to control for the different sleep measures as there were no specific hypotheses regarding sleep parameters. Significant or marginal Pearson correlation data were submitted to multiple regression analyses with simultaneous entry. Bootstrapping based on one thousand samples was conducted to control for test multiplicity. Depending on the Pearson correlation findings, independent variables (IVs) could be depression, social anxiety, and crime risk measures. Age was an IV regardless of the Pearson correlation findings as age differed between the diagnostic groups, and relationships between age and sleep have been reported [49,50]. The dependent variable (DV) was a sleep measure. For collinearity in the regression model to be satisfactory, tolerance was set at >0.30 [51,52]. To determine if significant relationships between IVs and the DV were distinct, post-hoc partial correlation analyses were conducted.

Prior to performing Pearson correlations and regression analyses, the Shapiro–Wilk test was used to test for normality. For variables that were not normally distributed, square root transformation was applied.

The Statistical Package for the Social Sciences (Chicago, Illinois version 27) was used to perform analyses and all analyses were two-tailed with the alpha level set to 0.05. 

## 3. Results

### 3.1. Participant Characteristics

As anticipated, independent-sample *t*-tests showed that participants with major depression (*n* = 24) were significantly more depressed (HAMD total score) than participants with social anxiety (*n* = 35) [*t*(57) = 2.856, *p* = 0.006]. Furthermore, participants with social anxiety were significantly more socially anxious (LSAS total score) than participants with major depression [*t*(57) = 9.978, *p* < 0.001]. On average, depression severity based on the HAMD total score was in the mild range for both diagnostic groups [27]. Conversely, the social anxiety disorder group exhibited marked social anxiety based on the LSAS total score, whereas the major depression group was in the range of mild social anxiety [42]. With regard to the actigraph device, the average number of days and nights the device was worn was 7.694 (*SD* = 1.949) and the average number of sleep logs completed was 79.7%. Independent sample *t*-tests showed the diagnostic groups did not significantly differ in the number of days the device was worn [*t*(57) = 0.448, *p* = 0.656]. However, there was a marginal effect for completion of sleep logs [χ^2^(1) = 3.599, *p* = 0.058], which appeared to be driven by the social anxiety group having fewer completed sleep logs (16.9%) relative to the major depression group (3.4%). To explore whether there were any differences between participants who completed the sleep logs (*n* = 47) and those who did not (*n* = 12), independent *t*-tests were performed. As may be expected given the previous results, the groups differed in their severity of social anxiety, where social anxiety was higher among participants who did not complete the sleep logs relative to those who completed the sleep logs [*t*(57) = 2.043, *p* = 0.046]. Yet, there was no difference in the depression level (HAMD total score), sleep quality (PSQI), actigraphic variables (TST, WASO, SOL), property crime risk, personal crime risk, or age (lowest *p* = 0.064). Chi-square tests also showed no differences between the groups with regard to ethnicity or race (lowest *p* = 0.346). 

Regarding sleep measures, sleep quality (PSQI) was not collected for one participant due to human error. No significant differences were detected between the depressed and social anxiety groups for actigraphic estimates of sleep (TST, WASO, SOL) or subjective sleep quality (PSQI) (lowest *p* = 0.141). As for crime risk, independent sample *t*-tests showed no significant differences between the diagnostic groups for personal [*t*(57) = 0.302, *p* = 0.764] or property crime risk [*t*(57) = 0.636, *p* = 0.528]. With regard to demographic characteristics, the average age in years across participants was 26.135 (*SD* = 8.669) and independent sample *t*-tests showed participants with major depression were significantly older (*M* = 28.875, *SD* = 9.633) than participants with social anxiety (*M* = 24.257, *SD* = 7.516) [*t*(57) = 2.066, *p* = 0.043]. As for self-identified gender, chi-square tests showed there were no significant difference between the diagnostic groups [χ^2^(2) = 0.845, *p* = 0.656]; one participant did not identify as a man or woman. Moreover, no significant differences between diagnostic groups were observed for self-identified ethnicity [χ^2^(1) = 0.104, *p* = 0.747] or race [χ^2^(5) = 3.765, *p* = 0.584]. See Table 3 for details. 

Accordingly, to aid in characterizing sleep in the current sample, we collapsed across diagnostic groups and report the averages and standard deviations (*SD*) for actigraphic variables as follows: TST = 6.88 h (*SD* = 0.98), range 4.56 to 9.22 h; WASO = 39.54 min (*SD* = 13.36), range 19.93 to 76.21 min; and SOL = 15.46 min (*SD* = 10.65), range 2.68 to 57.31 min. For subjective sleep quality (PSQI), the average global score was 7.931 (*SD* = 3.249), range 2.00 to 18.00, and the majority of participants (*n* = 44, 75.9%) met the criteria for clinically problematic sleep based on the PSQI global score cut-off point of greater than 5 [46]. Also, even though participants were not pre-selected for insomnia or hypersomnia, 27.1% (*n* = 16) met diagnostic criteria [41] for insomnia disorder and 8.5% (*n* = 5) met criteria for hypersomnia disorder. 

### 3.2. Normality Results

When evaluating the normality of measures for correlation and regression analyses, the Shapiro–Wilk test was not significant for depression (modified HAMD), social anxiety (LSAS), sleep quality (PSQI), and actigraphic TST (lowest *p* = 0.152), signifying a normal distribution. However, the Shapiro–Wilk test was significant for actigraphic WASO and SOL, in addition to personal and property crime risk and age (all *p*-values < 0.05). Therefore, square root transformation was applied to WASO, SOL, crime rate, and age variables for Pearson correlation and regression analyses. 

### 3.3. Pearson Correlation Results

Pearson correlations revealed a significant positive relationship between depression (modified HAMD) and sleep duration (actigraphic TST) (r = 0.356, *p* = 0.006), and a marginal positive relationship between depression (modified HAMD) and sleep quality (PSQI) (r = 0.295, *p* = 0.025) based on the Bonferroni correction (*p* = 0.025). No significant associations were observed between depression (modified HAMD) and actigraphic WASO or SOL (lowest *p* = 0.744). Also, no significant relationships were detected between social anxiety and any sleep measure (lowest *p* = 0.206). When evaluating crime risk variables, there was a significant positive relationship between personal crime risk and actigraphic SOL (r = 0.340, *p* = 0.008) but no other sleep measures (lowest *p* = 0.137). There was a non-significant trend of a positive relationship between property crime risk and actigraphic SOL (r = 0.257, *p* = 0.050), though the relationship was not close to significance when adjusting for multiple comparisons (*p* = 0.025). No other relationships between property crime risk and sleep were detected (lowest *p* = 0.193). 

### 3.4. Regression Results

Based on the Pearson correlation results, multiple regression analyses were performed with the following dependent variables (DV): actigraphic TST, actigraphic SOL, and sleep quality (PSQI). Independent variables (IVs) consisted of depression (modified HAMD), personal crime risk, and age. 

For the model with actigraphic TST as the DV, collinearity was satisfactory (tolerance > 0.30). Bootstrapped results revealed depression was positively related with TST (B = 0.102, *s.e*. = 0.037, *p *= 0.009) when taking personal crime risk and age into account. None of the other IVs were significant (lowest *p* = 0.091); the model was significant [*R*^2^ = 0.178, *F*(3,55) = 3.982, *p *= 0.012]. 

When actigraphic SOL was the DV, collinearity was satisfactory (tolerance > 0.30). Bootstrapped results showed personal crime risk was positively corresponded with SOL (B = 0.037, *s.e*. = 0.013, *p *= 0.005) and all other IVs were not significant (lowest *p* = 0.121); the model was significant [*R*^2^ = 0.145, *F*(3,55) = 3.097, *p *= 0.034]. 

With sleep quality (PSQI) as the DV, collinearity was satisfactory (tolerance > 0.30). Bootstrapped results revealed non-significant trends for depression (modified HAMD) (B = 0.246, *s.e*. = 0.127, *p *= 0.066) and age (B = 1.086, *s.e*. = 0.609, *p *= 0.058); personal crime risk was not significant (*p* = 0.248). The model was significant [*R*^2 ^= 0.180, *F*(3,54) = 3.962, *p *= 0.013]. 

Based on regression results, post-hoc correlational analyses were performed for depression (modified HAMD) and actigraphic TST, and personal crime risk and actigraphic SOL, as the relationships and models were significant. Partial correlation results showed that the depression–TST relationship was significant when controlling for total crime risk, which encompassed personal and property crime derived from the CrimeRisk database [48], social anxiety (LSAS), and age (r = 0.346, *p* = 0.009). Partial correlation results also showed the personal crime risk–SOL relationship was significant when controlling for symptom severity (modified HAMD, LSAS) and age (r = 0.346, *p* = 0.009). However, the personal crime risk–SOL relationship was not significant when controlling for property crime risk along with symptom severity and age (r = 0.221, *p* = 0.102). See Figure 1 for scatterplots for significant partial correlations. 

## 4. Discussion

The current study evaluated the extent to which internalizing symptoms (i.e., depression, social anxiety) and crime risk uniquely or jointly related to sleep in unmedicated treatment-seeking participants with major depressive disorder or social anxiety disorder. We expected greater symptom severity, that is, more depression and more social anxiety to be associated with worse sleep. We also hypothesized that greater neighborhood crime risk would be associated with worse sleep. Pearson correlation and regression findings suggest depression severity and crime risk differentially relate to sleep measures. There was no evidence of a relationship between social anxiety and sleep. Taken together, the hypotheses were partially supported. 

The optimal sleep duration is 7 to 9 h [53] and, on average, actigraphic estimates of sleep showed sleep duration in the current sample was 6.8 h a night with a range of 4.56 to 9.22 h. Sleep duration did not significantly differ between participants with major depression or social anxiety. Thus, the results indicate that many participants experienced insufficient sleep overall, though there was a considerable range in terms of sleep duration. While the upper number of hours of sleep may not be considered excessive, previous study findings point to a U-shaped association between sleep duration and depression such that both short and long sleep durations are linked with depression [54,55]. Therefore, the significant positive relationship observed between depression and actigraphic total sleep time may reflect mechanisms in keeping with ‘excessive’ sleep. For example, in depression, longer sleep duration may be due to difficulty coping with stress or negative mood [56] or low physical activity [57] that may have adverse direct or indirect effects [58,59]. Since we did not collect these data, it will be important for future studies to examine the mechanisms that underlie this relationship. Critically, the relationship between depression severity and sleep duration was maintained when adjusting for crime risk and age, suggesting that internal factors (e.g., depression) may be more relevant for sleep duration than an external signal of threat (i.e., neighborhood crime). In further support of a distinct depression–sleep duration relationship, the findings remained significant when performing a post-hoc correlational analysis controlling for crime rate, social anxiety, and age. 

In contrast to sleep duration, the findings revealed a higher crime risk corresponded with a longer actigraphic sleep onset latency when taking depression and age into account. Sleep onset latency represents the time it takes to enter sleep, which is typically less than 30 min [60,61] and in the current study, average sleep onset latency was 15 min, ranging from approximately 3 min to close to 60 min with no significant difference between participants with major depression or social anxiety. Therefore, the findings suggest that on average, there was no marked disturbance in sleep onset latency, though the range indicated some participants may have had difficulty falling asleep. Notably, zero-order correlations adjusting for multiple comparisons revealed a significant association between personal crime risk (i.e., murder, rape, robbery, assault) and sleep onset latency but not property crime risk (i.e., burglary, larceny, motor vehicle theft) and sleep onset latency. Though the findings would suggest that the ‘type’ of crime risk differentially relates to sleep onset latency, post-hoc correlation showed the significant personal crime risk–sleep onset latency relationship was not maintained when controlling for property crime. Thus, the findings indicate higher personal and property crime risks relate to longer sleep onset latency. With regard to potential mechanisms, we speculate that exposure to various crimes may contribute to longer sleep onset latency via negative thoughts and/or hyper-arousal among individuals with major depression or social anxiety. This is based on studies that have demonstrated that negative cognition (i.e., rumination) predicts longer sleep onset latency [62] and the adverse influence pre-sleep arousal has on sleep [17,20,21,22,23,24]. Of course, this will need to be tested in future studies before drawing such conclusions. 

As for actigraphic waking after sleep onset (WASO), an index of fragmented sleep, there were no significant differences between participants with major depression or social anxiety and Pearson correlations did not detect any significant relationships with symptom severity or crime risk. Consequently, a regression analysis was not performed with WASO as the dependent variable. It is not clear why individual differences in symptom severity and crime risk did not correspond with variance in WASO, and we hesitate to interpret null results. Replication in a larger, more heterogeneous sample is warranted. 

Regarding subjective sleep quality, the majority of participants (75.9%) met criteria for clinically problematic sleep (i.e., Pittsburgh Sleep Quality Index ‘PSQI’ global score greater than 5) [46] and there was no significant difference in sleep quality between participants with major depression or social anxiety. Interestingly, bootstrapped regression results showed non-significant positive depression–PSQI and age–PSQI trends when adjusting for crime risk, and the model was significant. We propose the significant omnibus test is due to the marginal effects of both depression and age as collinearity was acceptable. The depression finding is consistent with previous research. For example, positive relationships between depression severity and perceived poor sleep quality have been observed in major depressive disorder [63] and in the general adult population [64]. Similarly, the positive relationship with age is in keeping with reports that sleep quality tends to decrease with age [49,50]. The lack of a significant relationship between social anxiety and sleep quality suggests that fears of a social nature may not adversely affect sleep quality. Again, it will be important to replicate these results before firmly concluding that social anxiety does not impact subjective sleep quality.

In summary, preliminary findings indicate there were no significant differences in sleep between treatment-seeking participants with major depressive disorder or social anxiety disorder. Since the study did not include a healthy control comparator group, we cannot determine whether sleep was atypical. However, evidence that the majority of participants met criteria for clinically problematic sleep based on self-reporting suggests a substantial portion of participants experienced problematic sleep, yet diagnostic status had no effect. Therefore, when examining overall sleep, the results indicate actigraphic estimates of sleep and subjective sleep quality cut across diagnostic boundaries. Conversely, when evaluating individual differences, depression but not social anxiety severity corresponded with certain sleep measures, suggesting variance in depression may be more sensitive to sleep than variance in anxiety/fears involving social interaction or performance situations. Individual differences in crime risk also pertained to sleep, though findings were limited to sleep onset latency. Accordingly, it will be important for future studies to take a more fine-grained multifaceted approach to increase our understanding of these relationships. Also, the clinical inferences of these findings suggest individuals may benefit from a comprehensive approach when assessing sleep that encompasses neighborhood characteristics for more tailored treatment. 

### Limitations

Findings need to be considered in the context of important limitations. First, findings are vulnerable to type 1 error as associations with the various sleep measures were not adjusted for multiple comparisons, and the sample size was relatively small. Therefore, it will be important to replicate the study findings using a more rigorous approach in a larger sample. Second, we did not verify the length a participant resided at their location; thus, exposure to crime risk may have differed among participants due to moving. Also, objective neighborhood characteristics where an individual resides do not take into account other places an individual may spend considerable time at (e.g., work, school) [65] or perceived crime, which has been shown to differ from objective crime [66]. Third, other neighborhood factors that adversely affect sleep (e.g., unwanted noise, public drug use) [25] were not collected; it is possible that neighborhood factors outside of crime risk play a more prominent role in sleep among individuals with internalizing conditions. Fourth, depression, on average, was in the mild range in both diagnostic groups. Participants with major depressive disorder could not have comorbid social anxiety disorder and participants with social anxiety disorder could not have comorbid major depressive disorder; thus, the findings may not be generalizable to individuals with more severe depression or individuals with concurrent major depression and social anxiety. Relatedly, results may not be generalizable to individuals who are receiving pharmacotherapy or who differ in demographic or clinical characteristics from the current study population. Fifth, as there was no healthy comparator control group, it was not possible to determine whether estimates of sleep were atypical. Sixth, age significantly differed between diagnostic groups; therefore, we cannot rule out the possibility that age influenced the findings. Seventh, even though the actigraphic estimates of sleep onset latency are consistent with gold-standard polysomnography, individual differences may have low reliability [67]. Eighth, the majority of participants were Caucasian and non-Hispanic, which did not permit evaluation of the possible interactions between sleep and crime risk as it pertains to minority populations. Ninth, the cross-sectional design does not permit evaluation of symptom severity or crime risk as predictors of sleep. Tenth, we did not screen for certain sleep disorders (e.g., obstructive sleep apnea) or measure or instruct participants to refrain from activities (e.g., watching television while in bed) or substances (e.g., caffeine, alcohol) that may impact sleep, which may have introduced confounding factors. Eleventh, a portion of participants (i.e., 20.3%) did not complete a sleep log; thus, the actigraphy event marker press reflecting sleep onset/offset could not be verified in these participants. Also, we did not confirm that participants adhered to the instructions on completing sleep logs used for actigraphy analysis. Twelfth, there was no manipulation of sleep and the study was not designed to test mechanisms of sleep; therefore, the results rely on estimates of naturalistic sleep and are limited to associations among variables of interest. Lastly, data on participants’ weekday and weekend schedules or preferred times to engage in activities or sleep (e.g., chronotype) was not obtained; thus, we cannot rule out the potential that they may have influenced the findings. 

## 5. Conclusions

Preliminary results suggest overall actigraphic estimates of sleep and subjective sleep quality do not significantly differ between treatment-seeking participants with major depressive disorder or social anxiety disorder. The findings also suggest a substantial portion of participants with major depression or social anxiety experience clinically poor sleep quality. As for crime risk, no significant differences were detected between the diagnostic groups. When examining individual differences, the regression results indicate that higher depression scores correspond with longer actigraphic sleep duration when adjusting for crime risk and age, whereas a higher crime risk corresponds with longer actigraphic sleep onset latency when adjusting for depression and age. Accordingly, depression and crime risk relate differently to aspects of sleep. The clinical implications of these findings suggest it is important to consider sleep as a multifaceted construct and to assess neighborhood characteristics when evaluating sleep in clinical populations. 

## Figures and Tables

**Figure 1 brainsci-14-00104-f001:**
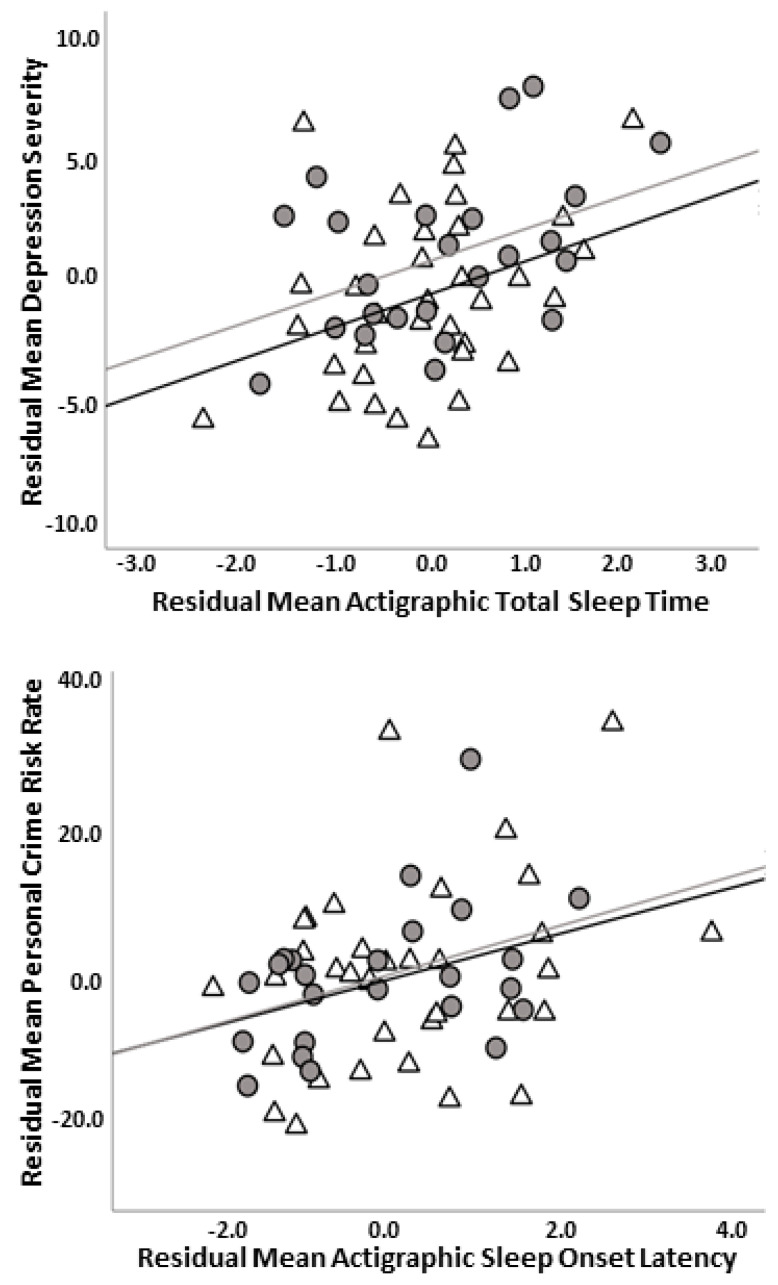
Scatterplot illustrating relationship between depression (modified HAMD) and actigraphic total sleep time controlling for crime risk, social anxiety (LSAS), and age in years (i.e., residuals) (top panel); scatterplot depicting relationship between personal crime risk (square root transformed) and actigraphic sleep onset latency (square root transformed) controlling for depression (modified HAMD), social anxiety (LSAS), and age in years (i.e., residuals) (bottom panel). Note: Modified HAMD = Hamilton Depression Rating Scale without insomnia items; LSAS = Liebowitz Social Anxiety Scale; gray circles and gray line = major depressive disorder group; white triangles and black line = social anxiety disorder group.

**Table 1 brainsci-14-00104-t001:** Psychiatric comorbidity as percentage for each diagnostic group.

	Major Depression Group (*n* = 24)	Social Anxiety Group (*n* = 35)
Comorbid Diagnoses (%)	%	%
Generalized Anxiety Disorder	45.8	45.7
Insomnia	37.5	20.0
Hypersomnolence	8.3	8.6
Specific Phobia	8.3	8.6
Persistent Depressive Disorder	29.2	0.0
Panic Disorder	0.0	2.9
Posttraumatic Stress Disorder	0.0	2.9
Attention-Deficit/Hyperactivity Disorder	0.0	2.9
Adjustment Disorder	0.0	2.9

**Table 2 brainsci-14-00104-t002:** Annual Illinois index crime offense rate for different crimes and year of crime.

Index Crime Rate	Homicide	Rape	Robbery	Aggravated Assault/Battery	Burglary	Theft	Motor Vehicle Theft
Year 2017	8.0	36.7	135.4	241.8	358.1	1454.2	150.3
Year 2018	6.9	45.2	110.0	234.3	292.9	1405.3	149.9
Year 2019	6.8	46.5	95.8	243.5	254.3	1355.1	143.2
Year 2020	9.2	37.9	94.5	261.5	227.7	1137.2	162.4

**Table 3 brainsci-14-00104-t003:** Clinical, sleep, crime risk, and demographic data and statistics for diagnostic groups: Values for the depressed and social anxiety groups reflect the mean, and standard deviations are in parentheses.

Measures	Major Depression Group (*n* = 24)	Social Anxiety Group (*n* = 35)	*t*-Value	*p*-Value
Hamilton Depression Rating Score	11.416 (3.866)	8.342 (4.186)	2.856	0.006
Liebowitz Social Anxiety Score	36.916 (16.557)	82.714 (17.813)	9.978	<0.001
Actigraphic total sleep time (hours)	6.861 (1.073)	6.888 (0.918)	0.104	0.917
Actigraphic waking after sleep onset (minutes)	38.334 (15.032)	40.372 (12.233)	0.572	0.569
Actigraphic sleep onset latency (minutes)	12.989 (8.880)	17.159 (11.525)	1.493	0.141
Pittsburgh Sleep Quality Index global score	8.500 (3.718)	7.529 (2.862)	1.123	0.266
Personal crime risk	672.416 (650.721)	733.971 (840.207)	0.302	0.764
Property crime risk	353.333 (214.798)	319.542 (190.411)	0.636	0.528
Age in years	28.875 (9.633)	24.257 (7.516)	2.066	0.043
Gender (%)	%	%	**Chi-square value**	***p*-Value**
			0.845	0.656
Women	66.7	60.0		
Men	33.3	37.1		
Not Reported	0.0	2.9		
Ethnicity (% Hispanic/Latinx)	16.7	20.0	0.104	0.747
Racial Identity (%)	%	%		
			3.765	0.584
White	45.8	54.3		
Black	16.7	17.1		
Asian	16.7	20.0		
Native Hawaiian or other Pacific Islander	0.0	0.0		
American Indian or Alaskan Native	0.0	0.0		
More than one race	8.3	2.9		
Other	12.5	2.9		
Not reported	0.0	2.9		

## Data Availability

The authors followed the National Institutes of Health (NIH) guidelines for data sharing, and data used for the original study aims are available through the NIH. Also, data and material that support the findings of the current study are available from the corresponding author upon reasonable request. The data are not publicly available due to confidentiality concerns and to ensure data integrity.

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
