# Peer review of "Crime Risk and Depression Differentially Relate to Aspects of Sleep in Patients with Major Depression or Social Anxiety"

_brainsci, 2024, doi:10.3390/brainsci14010104_

Round 1

Reviewer 1 Report

Comments and Suggestions for Authors

Thank you for the opportunity to review “Crime Risk and internalizing symptoms differentially relate with aspects of sleep in major depressive disorder and social anxiety disorder.” This is an interesting manuscript that utilized secondary data to understand how internalizing symptoms of depression and social anxiety and community crime uniquely influence sleep. There are a few areas that would benefit from revision prior to publication. I hope the reviewers find these comments helpful.

 Introduction 

The authors state in the introduction that longitudinal research has linked depression and insomnia, and established a link to sleep disturbances with anxiety. They further suggest problematic sleep contributes to the development and exacerbation of depression and anxiety and may be trans- diagnostic. They also state that community crime is connected with poor or insufficient sleep and make the connections to depression/anxiety might interact with community crime to contribute to sleep difficulties, suggesting this has been understudied. Based on the introduction the premise is that sleep is related to diagnoses of MDD and anxiety, however the rationale of the manuscript focused on internalizing symptoms and only one specific SAD. Further expanding on the shift away from diagnosis to symptoms would be helpful to establish a clear understanding to the objective of the manuscript. It was understood the authors hypothesized more symptoms, more sleep problems, but having more to support this assertion would be useful to the reader. In addition, the discussion leads off with “the current study evaluated the extent to which internalizing symptom severity and crime risk correspond with actigraphic estimates of sleep….” This seems a bit divergent from the initial rationale of looking at internalizing symptoms and crime risk. Based on this statement, it appears that the authors are more interested in the severity of diagnosis vs. specific internalizing symptoms. The framing of the manuscript is at times confusing and reworking the frame and consistent presentation would help the readability of the manuscript.  

Research Design

There are a few areas that were not clear and might be influencing the results. Some things to consider:

-       The authors removed items 4 – 7 on the HAMD, how did this impact the overall reliability of the measure? Was there a reason to not include reliability data? It would seem that the sensitivity of discriminating severity of symptoms would have been impacted with the removal of these items (see Carrozzino et al, 2020). Also it seems the mild symptoms for the sample might be accounted for due to the removal of these items. Just something to consider.

-       Some more detail on the neighborhood crime exposure measure calculation would be helpful. It was clear the mechanism used was a crime risk data base that was analyzed over a 7 year period. Was there variability over the 7 year period? It seems crime can be variable and it would be useful to understand if there was much change over time. There could have been a lot of crime in 2015, not so much for 2023 and the participant may have only lived there for a year. I know time living in a neighborhood was mentioned in the limitations, but based on what is outlined, relative risk is not current exposure. Maybe, this is in error, but overall how this measure and data was determined was a bit confusing. As currently written there seems like there could be significant limitations to a key variable in the manuscript. Also, it appears this is zip code data within the county. Zip codes can be large and have variability of risk for crime. It seems possible that while some may reside in a zip code with high crime, their particular block might be insulated from crime for various reasons. Addressing this a bit more in the results and or limitations would be helpful.

-       Based on the stated objective of interest in internalizing symptomology, it was not clear as the reason for using the total scale scores if the goal was to look at only internalizing symptoms. Each measure has diversity in indicators, thus there seems to be a mis-match in goals/objectives. Again, this might just be related to the framing of the intro etc.  

-       The LSAS appears to measure various aspects of social anxiety beyond internalizing, again just confusing in the framing and goals/objectives.

-       Just a small note, the authors used insomnia in table 1 as a comorbid condition, wouldn’t this be an outcome?

 Minor edits

Just a minor note, there were multiple times that the font was changes or there was bold/underline. Not sure if this was on the pdf conversion, just wanted to mention it as an FYI.

Author Response

Thank you for the opportunity to review “Crime Risk and internalizing symptoms differentially relate with aspects of sleep in major depressive disorder and social anxiety disorder.” This is an interesting manuscript that utilized secondary data to understand how internalizing symptoms of depression and social anxiety and community crime uniquely influence sleep. There are a few areas that would benefit from revision prior to publication. I hope the reviewers find these comments helpful.

Introduction 

The authors state in the introduction that longitudinal research has linked depression and insomnia, and established a link to sleep disturbances with anxiety. They further suggest problematic sleep contributes to the development and exacerbation of depression and anxiety and may be trans- diagnostic. They also state that community crime is connected with poor or insufficient sleep and make the connections to depression/anxiety might interact with community crime to contribute to sleep difficulties, suggesting this has been understudied. Based on the introduction the premise is that sleep is related to diagnoses of MDD and anxiety, however the rationale of the manuscript focused on internalizing symptoms and only one specific SAD. Further expanding on the shift away from diagnosis to symptoms would be helpful to establish a clear understanding to the objective of the manuscript. It was understood the authors hypothesized more symptoms, more sleep problems, but having more to support this assertion would be useful to the reader. In addition, the discussion leads off with “the current study evaluated the extent to which internalizing symptom severity and crime risk correspond with actigraphic estimates of sleep….” This seems a bit divergent from the initial rationale of looking at internalizing symptoms and crime risk. Based on this statement, it appears that the authors are more interested in the severity of diagnosis vs. specific internalizing symptoms. The framing of the manuscript is at times confusing and reworking the frame and consistent presentation would help the readability of the manuscript.  

Response:  We agree with the Reviewer and appreciate the opportunity to clarify.  Using a conventional diagnostic framework, both major depressive disorder (MDD) and social anxiety disorder (SAD) are internalizing psychopathologies (e.g., Friedman et al., 2011; Andrews 2018).  Therefore, MDD and SAD represent internalizing disorders and thus depression and social anxiety measures (i.e., HAMD, LSAS) reflect internalizing symptoms. To increase the clarity of the study, we have made the following modifications.

The title now uses ‘depression’ instead of ‘internalizing.’

Page 1, starting Line 36: “Altogether, there is evidence problematic sleep is pervasive and may be a transdiagnostic feature of depressive and anxiety disorders as it cuts across diagnostic categories.”

Page 2, starting Line 67:  “According to social-ecological models of sleep health, individual level (e.g., mental health) and neighborhood factors (e.g., neighborhood safety) contribute to sleep difficulties11,31.  However, the extent to which internalizing symptoms (i.e., depression, anxiety) and neighborhood crime relate to sleep uniquely or together remains unclear as this has been under-researched.” 

Page 2, starting Line 84:  “Based on previous study findings that indicate sleep difficulties are transdiagnostic in that they have been observed in both depressive and anxiety disorders together with evidence greater depression severity and social anxiety severity are associated with more problematic sleep6,39–41, we hypothesized higher levels of depression and higher levels of social anxiety would correspond with worse sleep (e.g., shorter sleep duration, worse sleep quality).”   

In the Discussion on Page 9, starting Line 352, we now state:  “The current study evaluated the extent to which internalizing symptoms (i.e., depression, social anxiety) and crime risk uniquely or jointly related to sleep in unmedicated treatment-seeking participants with major depressive disorder or social anxiety disorder.” 

Research Design

There are a few areas that were not clear and might be influencing the results. Some things to consider:

-       The authors removed items 4 – 7 on the HAMD, how did this impact the overall reliability of the measure? Was there a reason to not include reliability data? It would seem that the sensitivity of discriminating severity of symptoms would have been impacted with the removal of these items (see Carrozzino et al, 2020). Also it seems the mild symptoms for the sample might be accounted for due to the removal of these items. Just something to consider.

Response:  We agree and now report internal consistency for the total depression measure (HAMD) and modified depression measure (HAMD), namely, removing items 4, 5, and 6.  For consistency, we also report internal consistency for the social anxiety measure (LSAS).  Findings both the HAMD and modified HAMD are similar in their internal consistency as follows:

Page 4, starting Line 152:  “Internal consistency for the HAMD in the current study was acceptable (Cronbach’s alpha = 0.648)45,46. When evaluating relationships between depression severity and sleep measures, a modified version of the HAMD was used.  Specifically, Items 4, 5, and 6 were removed as they assess early, middle, and late insomnia, respectively.  This version of the HAMD is hereafter referred to as the ‘modified HAMD’. Internal consistency for the modified HAMD was also acceptable (Cronbach’s alpha = 0.623) as Cronbach’s alpha was between 0.60 and 0.7045,46

Response:  As for the mild symptoms of depression in the sample, we clarify that it was based on the HAMD total score. On Page 6, starting Line 244, we report: “On average, depression severity based on the HAMD total score was in the mild range for both diagnostic groups28.”

-       Some more detail on the neighborhood crime exposure measure calculation would be helpful. It was clear the mechanism used was a crime risk data base that was analyzed over a 7 year period. Was there variability over the 7 year period? It seems crime can be variable and it would be useful to understand if there was much change over time. There could have been a lot of crime in 2015, not so much for 2023 and the participant may have only lived there for a year. I know time living in a neighborhood was mentioned in the limitations, but based on what is outlined, relative risk is not current exposure. Maybe, this is in error, but overall how this measure and data was determined was a bit confusing. As currently written there seems like there could be significant limitations to a key variable in the manuscript. Also, it appears this is zip code data within the county. Zip codes can be large and have variability of risk for crime. It seems possible that while some may reside in a zip code with high crime, their particular block might be insulated from crime for various reasons. Addressing this a bit more in the results and or limitations would be helpful.

Response:  We agree with the Reviewer’s points.  First, we apologize that the version reviewed did not include important feedback by a co-author in our original manuscript concerning how neighborhood crime exposure was estimated and our subsequent attempts to have the version with the co-author’s feedback reviewed were unsuccessful.  Therefore, among the modifications, we report that the participant’s residential address was used to estimate neighborhood crime, as opposed to just zip code.

Page 4, starting Line 182:  “The 2023 CrimeRisk database49 was used to obtain indices of neighborhood crime risk at the block group level from participants who resided in Illinois.  The database is derived from analysis of past crime reports and provides an assessment of the relative risk of personal (i.e., murder, rape, robbery, assault) and property (i.e., burglary, larceny, motor vehicle theft) crime occurring in one’s neighborhood compared to the national average. Participants’ neighborhoods were defined as the census block group of their current home address at the time of study enrollment.”  

Second, we have added estimates of crime rate in Illinois, where participants resided, for each year data was collected in Table 2. 

Page 5, starting Line 201:

“2.2.4. Crime in Illinois

To estimate the rates of crimes more broadly during the data collection period from 2017 to 2020, inclusive, we report the annual index crime reports from the Illinois State Police for these years (Table 2) since participants in the current study resided in Illinois. The index crime report reflects offenses, arrests, and associated crime rates per 100,000 inhabitants; categories of crime include criminal homicide, rape, robbery, aggravated battery/aggravated assault, burglary, theft, and motor vehicle theft among other crimes (e.g., arson).”

Table 2. Annual Illinois index crime offense rate for different crimes and year of crime.

Index Crime Rate

Homicide

Rape

Robbery

Aggravated

Assault/Battery

Burglary

Theft

Motor Vehicle Theft

Year 2017

8.0

36.7

135.4

241.8

358.1

1454.2

150.3

Year 2018

6.9

45.2

110.0

234.3

292.9

1405.3

149.9

Year 2019

6.8

46.5

95.8

243.5

254.3

1355.1

143.2

Year 2020

9.2

37.9

94.5

261.5

227.7

1137.2

162.4

Third, we have added the following under limitations in 4.1. Limitations.

Page 11, starting Line 448: “Also, objective neighborhood characteristics where an individual resides does not take into account other places an individual may spend considerable time at (e.g., work, school)66 or perceived crime, which has been shown to differ from objective crime67.”

-       Based on the stated objective of interest in internalizing symptomology, it was not clear as the reason for using the total scale scores if the goal was to look at only internalizing symptoms. Each measure has diversity in indicators, thus there seems to be a mis-match in goals/objectives. Again, this might just be related to the framing of the intro etc.  

Response:  The Reviewer raises an interesting point.  The rationale for using the total scale scores for depression and social anxiety is to increase the stability of findings.  As noted in our previous response to the Reviewer, we now report the internal consistency for depression and social anxiety measures.  We propose that evaluating each symptom separately (e.g., sadness, anhedonia, social anxiety for public speaking, social anxiety for meeting strangers) would reduce the ability to replicate findings.  Also, using total scores in regression analyses is consistent with our previous studies, for example, a recent study that evaluated sleep and suicidal ideation (Klumpp et al., Brain Sciences, 13(2):288, 2023).

-       The LSAS appears to measure various aspects of social anxiety beyond internalizing, again just confusing in the framing and goals/objectives.

Response:  We apologize for the confusion.  As stated on Page 2, Line 76, “Common symptoms of major depression include sadness, loss of pleasure, problems with appetite, and dysregulated activity level (e.g., psychomotor retardation) whereas social anxiety is marked by excessive anxiety and/or avoidance concerning different social and performance situations such as meeting strangers and public speaking27.”  Thus, the LSAS evaluates both fear and avoidance.  Since the LSAS assesses social anxiety, an internalizing disorder, the LSAS is limited to internalizing symptoms.  Put another way, fear disorders (e.g., social anxiety disorder, specific phobia, panic disorder) are considered to be internalizing conditions (e.g., Friedman et al., 2011; Andrews 2018), which often involve an avoidance component (e.g., avoiding strangers due to anxiety/fear of strangers).  Therefore, measures that evaluate anxiety, fear, and avoidance are in line with internalizing symptoms.

-       Just a small note, the authors used insomnia in table 1 as a comorbid condition, wouldn’t this be an outcome?

Response:  We had made the following modifications to clarify insomnia as a comorbid condition.  Also, relatively few participants met diagnostic criteria for insomnia disorder, therefore, using it as an outcome variable would involve a relatively unbalanced dataset.   

Page 3, starting Line 119:  See Table 1 for comorbid diagnoses, which includes insomnia and hypersomnolence. Since the original study did not have sleep aims, a comorbid sleep disorder was neither an inclusion or exclusion criterion.

 Minor edits

Just a minor note, there were multiple times that the font was changes or there was bold/underline. Not sure if this was on the pdf conversion, just wanted to mention it as an FYI.

Response:  We thank the Reviewer for bringing this to our attention and believe these errors were due to conversion after loading the manuscript in the Journal portal.  We have made the corrections and hope they will be maintained.

In this manuscript, Klumpp et al. examine risks for crime and internalizing symptoms differentially relate to sleep parameters in patients with major depressive disorders (MDD) and social anxiety disorder (SAD). They examined a sample of 24 participants with MDD and 35 with SAD and acquired both demographics and actigraphy measures of sleep and neighborhood-level risks for personal and property crime. They found that the groups did not differ in sleep or neighborhood crime indices. However, they also found that crime risk correlated positively with sleep onset latency and that depression correlated with total sleep time. The authors suggest that crime exposure and depression rate differentially to sleep parameters in the patient groups. The paper is generally well-written and covers an important and understudied problem. However, several points need to be addressed.

It appears that the participants were not medicated. This should be more clearly stated in the description of participants.

Response:  We agree and now report on Page 2, Line 97 that participants are unmedicated as follows: “The original study used a multimethod approach to determine predictors of psychotherapy outcome and mechanisms of symptom improvement following psychotherapy in unmedicated participants seeking treatment for major depression or social anxiety  (ClinicalTrials.gov Identifier: NCT03175068).” 

The main problem in the manuscript is that the issue of multiple comparisons is not clear. Correlations were not corrected for multiple comparisons. Variables that showed significant correlations were entered as independent variables into a simultaneous multiple regression to "evaluate the stability of results." These used bootstrapping. It appears that this step was intended to control for test multiplicity, but this is not made clear. These are conducted with sleep parameters as the dependent variables (in separate regressions). It should be made more explicit how specifically the multiple comparison was handled.

Response:  We agree with the Reviewer and now report on Page 5, starting Line 214 the approach to correct for multiple comparisons as follows: “To evaluate relationships between internalizing symptoms (i.e., depression, social anxiety) and sleep in addition to crime risk and sleep, a series of Pearson correlations were performed.  To adjust for multiple comparisons, Bonferroni correction was applied for the two symptom measures (i.e., depression, social anxiety) and the two crime risk measures (personal, property).  Specifically, when evaluating depression-sleep relationships and social anxiety-sleep relationships, p=0.025 (0.05/2) denoted significance.  The same approach was used when evaluating personal and property crime risk as they pertain to sleep.  In light of the novelty of the study, we elected not to control for the different sleep measures as there were no specific hypotheses regarding sleep parameters.”  We also state the following under limitations on Page 11 starting Line 442:  “First, findings are vulnerable for Type 1 error as associations with the various sleep measures were not adjusted for multiple comparisons, and the sample size was relatively small.”    

In the results section, a nonsignificantly higher percentage of the SAD group failed to complete diaries than did the MDD group. Were there any differences between those who did or did not complete diaries in any of the other measures? Moreover, none of the data from the diaries is presented. They should be presented or the reason for not reporting this data should be given.

Response:  We agree with the Reviewer and now report analyses testing for differences between participants who completed the diaries and those who did not on Page 6, starting Line 254:

“To explore whether there were any differences between participants who completed the sleep logs (n=47) and those who did not (n=12), independent t-tests were performed.  As may be expected given previous results, the groups differed in severity of social anxiety where social anxiety was higher among participants who did not complete the sleep logs relative to those who completed the sleep logs [t(57)=2.043, p=0.046].  Yet, there was no difference in depression level (HAMD total score), sleep quality (PSQI), actigraphic variables (TST, WASO, SOL), property crime risk, personal crime risk, or age (lowest p=0.064).  Chi-square tests also showed no differences between the groups with regard to ethnicity or race (lowest p=0.346).”

Response: We have also added more information regarding the sleep diaries on Page 4, starting Line 163 and rationale for not analyzing the sleep diaries:

“They were also instructed to press the event marker on the device when intending to go to sleep and waking up and to record sleep and wake times in a sleep log, which was used for actigraphy data processing.  The sleep log also consisted of binary responses (i.e., yes/no) to three questions that related to sleep problems (e.g., “Did you have difficulty staying asleep?”, “Do you think you have a sleep problem?”).  There was also a question regarding removal of the wrist actigraphy device and if it was removed, the duration it was not worn.  Since the primary goal of the sleep log was for processing actigraphy data, responses to the three sleep questions were not entered for analysis in the current study.”

We also report here that binary responses to the three sleep-related questions would have restricted range, which reduces the potential for significant results.

Inferential statistics should be added into Table 2.

Response: We have added t values, chi-square values, and p values to Table 2 (now Table 3).

I found the discussion of the specificity of the crime type to be confusing (2nd full paragraph on page 9).  In particular, on lines 336-338, the authors state that both personal and property crime risk was related to sleep onset latency. However, the result in the regression model was only significant when personal crime risk was the IV. In lines 344-347, the authors state that ". . . the crime risk-sleep onset latency relationship was not unique to type of crime." But they then say "significant results were not maintained when controlling for property crime risk as well." Moreover, on lines 290-91, they state that "the personal crime risk-SOL relationship was not significant when controlling for property crime . . ." It thus appear that zero order correlations are significant for both types of crime, but the results for personal crime do not hold up when controlling for property crime. This should be clarified and its implications for their results further explicated.

Response:  We agree with the Reviewer and modified this section of the Discussion.  Specifically, on Page 10 starting Line 387, we now state:

“Notably, zero order correlations adjusting for multiple comparisons revealed a significant association between personal crime risk (i.e., murder, rape, robbery, assault) and sleep onset latency but not property crime risk (i.e., burglary, larceny, motor vehicle theft) and sleep onset latency.  Though findings would suggest ‘type’ of crime risk differentially relates with sleep onset latency, post-hoc correlation showed the significant personal crime risk-sleep onset latency relationship was not maintained when controlling for property crime.  Thus, findings indicate more personal and property crime risk relate to longer sleep onset latency.”   

For page 2, line 84, I suggest changing "sleep is" to "sleep problems are"

Response:  We agree and have made this change.

Reviewer 2 Report

Comments and Suggestions for Authors

In this manuscript, Klumpp et al. examine risks for crime and internalizing symptoms differentially relate to sleep parameters in patients with major depressive disorders (MDD) and social anxiety disorder (SAD). They examined a sample of 24 participants with MDD and 35 with SAD and acquired both demographics and actigraphy measures of sleep and neighborhood-level risks for personal and property crime. They found that the groups did not differ in sleep or neighborhood crime indices. However, they also found that crime risk correlated positively with sleep onset latency and that depression correlated with total sleep time. The authors suggest that crime exposure and depression rate differentially to sleep parameters in the patient groups. The paper is generally well-written and covers an important and understudied problem. However, several points need to be addressed.

It appears that the participants were not medicated. This should be more clearly stated in the description of participants.

The main problem in the manuscript is that the issue of multiple comparisons is not clear. Correlations were not corrected for multiple comparisons. Variables that showed significant correlations were entered as independent variables into a simultaneous multiple regression to "evaluate the stability of results." These used bootstrapping. It appears that this step was intended to control for test multiplicity, but this is not made clear. These are conducted with sleep parameters as the dependent variables (in separate regressions). It should be made more explicit how specifically the multiple comparison was handled.

In the results section, a nonsignificantly higher percentage of the SAD group failed to complete diaries than did the MDD group. Were there any differences between those who did or did not complete diaries in any of the other measures? Moreover, none of the data from the diaries is presented. They should be presented or the reason for not reporting this data should be given.

Inferential statistics should be added into Table 2.

I found the discussion of the specificity of the crime type to be confusing (2nd full paragraph on page 9).  In particular, on lines 336-338, the authors state that both personal and property crime risk was related to sleep onset latency. However, the result in the regression model was only significant when personal crime risk was the IV. In lines 344-347, the authors state that ". . . the crime risk-sleep onset latency relationship was not unique to type of crime." But they then say "significant results were not maintained when controlling for property crime risk as well." Moreover, on lines 290-91, they state that "the personal crime risk-SOL relationship was not significant when controlling for property crime . . ." It thus appear that zero order correlations are significant for both types of crime, but the results for personal crime do not hold up when controlling for property crime. This should be clarified and its implications for their results further explicated.

For page 2, line 84, I suggest changing "sleep is" to "sleep problems are"

Author Response

In this manuscript, Klumpp et al. examine risks for crime and internalizing symptoms differentially relate to sleep parameters in patients with major depressive disorders (MDD) and social anxiety disorder (SAD). They examined a sample of 24 participants with MDD and 35 with SAD and acquired both demographics and actigraphy measures of sleep and neighborhood-level risks for personal and property crime. They found that the groups did not differ in sleep or neighborhood crime indices. However, they also found that crime risk correlated positively with sleep onset latency and that depression correlated with total sleep time. The authors suggest that crime exposure and depression rate differentially to sleep parameters in the patient groups. The paper is generally well-written and covers an important and understudied problem. However, several points need to be addressed.

It appears that the participants were not medicated. This should be more clearly stated in the description of participants.

Response:  We agree and now report on Page 2, Line 97 that participants are unmedicated as follows: “The original study used a multimethod approach to determine predictors of psychotherapy outcome and mechanisms of symptom improvement following psychotherapy in unmedicated participants seeking treatment for major depression or social anxiety  (ClinicalTrials.gov Identifier: NCT03175068).” 

The main problem in the manuscript is that the issue of multiple comparisons is not clear. Correlations were not corrected for multiple comparisons. Variables that showed significant correlations were entered as independent variables into a simultaneous multiple regression to "evaluate the stability of results." These used bootstrapping. It appears that this step was intended to control for test multiplicity, but this is not made clear. These are conducted with sleep parameters as the dependent variables (in separate regressions). It should be made more explicit how specifically the multiple comparison was handled.

Response:  We agree with the Reviewer and now report on Page 5, starting Line 214 the approach to correct for multiple comparisons as follows: “To evaluate relationships between internalizing symptoms (i.e., depression, social anxiety) and sleep in addition to crime risk and sleep, a series of Pearson correlations were performed.  To adjust for multiple comparisons, Bonferroni correction was applied for the two symptom measures (i.e., depression, social anxiety) and the two crime risk measures (personal, property).  Specifically, when evaluating depression-sleep relationships and social anxiety-sleep relationships, p=0.025 (0.05/2) denoted significance.  The same approach was used when evaluating personal and property crime risk as they pertain to sleep.  In light of the novelty of the study, we elected not to control for the different sleep measures as there were no specific hypotheses regarding sleep parameters.”  We also state the following under limitations on Page 11 starting Line 442:  “First, findings are vulnerable for Type 1 error as associations with the various sleep measures were not adjusted for multiple comparisons, and the sample size was relatively small.”    

In the results section, a nonsignificantly higher percentage of the SAD group failed to complete diaries than did the MDD group. Were there any differences between those who did or did not complete diaries in any of the other measures? Moreover, none of the data from the diaries is presented. They should be presented or the reason for not reporting this data should be given.

Response:  We agree with the Reviewer and now report analyses testing for differences between participants who completed the diaries and those who did not on Page 6, starting Line 254:

“To explore whether there were any differences between participants who completed the sleep logs (n=47) and those who did not (n=12), independent t-tests were performed.  As may be expected given previous results, the groups differed in severity of social anxiety where social anxiety was higher among participants who did not complete the sleep logs relative to those who completed the sleep logs [t(57)=2.043, p=0.046].  Yet, there was no difference in depression level (HAMD total score), sleep quality (PSQI), actigraphic variables (TST, WASO, SOL), property crime risk, personal crime risk, or age (lowest p=0.064).  Chi-square tests also showed no differences between the groups with regard to ethnicity or race (lowest p=0.346).”

Response: We have also added more information regarding the sleep diaries on Page 4, starting Line 163 and rationale for not analyzing the sleep diaries:

“They were also instructed to press the event marker on the device when intending to go to sleep and waking up and to record sleep and wake times in a sleep log, which was used for actigraphy data processing.  The sleep log also consisted of binary responses (i.e., yes/no) to three questions that related to sleep problems (e.g., “Did you have difficulty staying asleep?”, “Do you think you have a sleep problem?”).  There was also a question regarding removal of the wrist actigraphy device and if it was removed, the duration it was not worn.  Since the primary goal of the sleep log was for processing actigraphy data, responses to the three sleep questions were not entered for analysis in the current study.”

We also report here that binary responses to the three sleep-related questions would have restricted range, which reduces the potential for significant results.

Inferential statistics should be added into Table 2.

Response: We have added t values, chi-square values, and p values to Table 2 (now Table 3).

I found the discussion of the specificity of the crime type to be confusing (2nd full paragraph on page 9).  In particular, on lines 336-338, the authors state that both personal and property crime risk was related to sleep onset latency. However, the result in the regression model was only significant when personal crime risk was the IV. In lines 344-347, the authors state that ". . . the crime risk-sleep onset latency relationship was not unique to type of crime." But they then say "significant results were not maintained when controlling for property crime risk as well." Moreover, on lines 290-91, they state that "the personal crime risk-SOL relationship was not significant when controlling for property crime . . ." It thus appear that zero order correlations are significant for both types of crime, but the results for personal crime do not hold up when controlling for property crime. This should be clarified and its implications for their results further explicated.

Response:  We agree with the Reviewer and modified this section of the Discussion.  Specifically, on Page 10 starting Line 387, we now state:

“Notably, zero order correlations adjusting for multiple comparisons revealed a significant association between personal crime risk (i.e., murder, rape, robbery, assault) and sleep onset latency but not property crime risk (i.e., burglary, larceny, motor vehicle theft) and sleep onset latency.  Though findings would suggest ‘type’ of crime risk differentially relates with sleep onset latency, post-hoc correlation showed the significant personal crime risk-sleep onset latency relationship was not maintained when controlling for property crime.  Thus, findings indicate more personal and property crime risk relate to longer sleep onset latency.”   

For page 2, line 84, I suggest changing "sleep is" to "sleep problems are"

Response:  We agree and have made this change.

Round 2

Reviewer 2 Report

Comments and Suggestions for Authors

This revision is a significant improvement over the earlier submission. I do not have additional comments other than to say that the authors may wish to state that use of multiple regression analyses with bootstrapping may provide additional protection over Type I error (I'm not an expert in this area, but I think that is correct).